# Early environments and exploration in the preschool years

Ilona Bass[1,2]*, Elizabeth Bonawitz[1]

**1** Graduate School of Education, Harvard University, Cambridge, Massachusetts, United States of America
**2** Department of Psychology, Harvard University, Cambridge, Massachusetts, United States of America

* ibass@fas.harvard.edu

**Data Availability Statement:** De-identified data and analysis scripts are publicly available in our OSF repository for this project: https://osf.io/gdqnk/?view_only=e69b227682143e79f9847a416b2e103.

## Abstract

A great deal of research has demonstrated how children's exploration is driven by opportunities for learning. However, less work has investigated how individual differences across children and their environmental contexts relate to patterns in playful exploration. We performed a "mega-analysis" in which we pooled preschool-aged children's play data from four past experiments in our lab (N = 278; $M_{(age)}$ = 56 months) and correlated play behaviors with age and socioeconomic status (median income, modal education in children's home zip codes). We found that, with age, children performed more unique actions during play. Additionally, children from lower SES areas explored more variably; the link between this play and tendencies to focus on pedagogically demonstrated features traded off differently than it did for higher SES children. This work lays critical groundwork for understanding exploration across developmental contexts.

## Introduction

Children learn by exploring the world. Past research has found a great deal of variation in how parents play with their children [1], and that the degree and quality of instruction provided by adults can shape the ways in which children subsequently explore [2–5]. Such *directed play* (i.e., independent exploratory play following a single piece of information provided by an adult) is different from other kinds of exploratory learning that have been the focus of past work. For example, *guided play* involves continuous scaffolding and feedback from adults that children can use to adjust their exploratory behavior in real-time [6–8]. Directed play, on the other hand, involves only a single piece of initial pedagogy, after which children are left to explore independently. Studying these behaviors provides a unique opportunity to understand how individual pieces of information provided by adults could shape child-driven learning, while also providing clearer insight into children's expectations and inferences in pedagogical contexts [2].

Directed play has been shown to be related to some early childhood experiences; for instance, children whose parents use more questioning as opposed to directives tend to explore in more efficient and variable ways [9]. But broadly, we know relatively little about how individual differences and children's early experiences are related to the ways in which they explore

**Funding:** This work was funded by an Early Career Research Fellowship from the Jacobs Foundation to EB (https://jacobsfoundation.org/), and a Scholar Award in Understanding Human Cognition from the James S. McDonnell Foundation to EB (https://www.jsmf.org/). IB is supported by a Mind Brain Behavior Postdoctoral Fellowship from Harvard University (https://mbb.harvard.edu/) and a STEM Education Postdoctoral Research Fellowship from the National Science Foundation (Award Number: 227447). There was no additional external funding received for this study. The funders had no role in study design, data collection and analysis, decision to publish, or preparation of the manuscript.

**Competing interests:** The authors have declared that no competing interests exist.

following directed play prompts. In the current work, we take a "mega-analytic" approach in which we compile and normalize several existing datasets from our lab to explore age-related differences in play following such direct prompts.

A second measure of interest concerns the relationship of environmental factors and play. Because our approach averages over past datasets, we are unable to directly measure relevant environmental factors like parenting styles, or past home and school experiences, which were not collected in the original studies. However, we are able to investigate whether home socio-economic status (SES) is related to patterns in preschool-aged children's directed play. Individual variation in factors like parenting styles and daycare settings tend to be related to differences in SES [10–14], so outcomes related to SES may reveal promising future pathways for direct investigation of the relevant environmental factors.

## Relating play to environmental factors

At least as early as Piaget [15], it has been suggested that children are active learners, in that they gather evidence for themselves by playing and exploring the world around them [16–18]. But children do not simply play in isolation, and not all exploration is created equal. Many factors may shape the ways in which children explore, including potential opportunities for learning [16, 19–21], prior success on unrelated tasks [22], and how information is presented and framed by other people [2, 3, 5]. At the group level, then, exploratory play may be influenced by various elements in children's environments–both external (e.g., pedagogical contexts) and internal (e.g., feelings of competence), as well as the interplay between the two (e.g., how events are interpreted with respect to the child's current beliefs). Less clear, however, is how exploratory play may systematically differ between individual children. While many have argued that environmental variables may play crucial roles in children's cognitive development [23], we do not fully understand the extent to which directed play may be related to these kinds of factors. In order to understand how to best promote early learning for children from a breadth of backgrounds–a goal of educators and developmentalists alike–it will be critical to probe these questions.

Past findings suggest that children's exploration might be meaningfully related to factors such as parenting style [9, 24, 25] or environmental reliability [26]. One recent study investigated the relationships between cumulative risk (comprising factors such as family income and maternal stress), children's ability to stay on-task in a puzzle-solving activity, and the amount of scaffolding versus directives provided by mothers during this task [27]. Results revealed that higher cumulative risk was broadly associated with decreased use of scaffolding; and when children were on-task, high-risk mothers were more likely to provide directives than low-risk mothers were [27]. Separate studies have found that children tend to explore more broadly and discover more on their own following instruction in the form of pedagogical questions (one type of scaffolding behavior), as compared with directives [5]. Parents from families with lower SES tend to ask their children fewer pedagogical questions at home [25] and children whose parents ask fewer pedagogical questions also tend to explore less in a subsequent, independent novel toy task following a pedagogical prompt [9], indicating a relationship between past home experiences and inferred play behaviors. Together, these findings suggest that early home experiences predict differences in children's directed play, which may be proximally captured by SES.

In addition to being a reasonable proxy for the early experiences in which we are interested, familial SES is itself tied to later outcomes for children, suggesting this may be a particularly important variable to examine as it relates to children's early exploratory behaviors. Early emerging disparities in school achievement between children from high- and low-SES

backgrounds have been documented in past work; for a meta-analysis, see [28]. Given that playful learning approaches have been found to be related to later school readiness [29], we believe it is important to characterize these relationships. To be clear, it is the factors that tend to correlate with SES, and not SES itself, that are likely the important causal variables in shaping children's expectations in pedagogically cued play. Nonetheless, it could be useful to examine how SES is related to patterns in children's exploration–particularly whether pedagogical cues shape directed play differently across individual children from different backgrounds.

## Relating play to development

In addition to environmental factors, it may also be informative to investigate how directed play changes with age within the preschool years. Some of the cognitive capacities that could support productive exploration, such as inhibitory control, planning, and goal-switching, develop rapidly during early childhood [30]. Recent work has also found that the breadth and efficiency of at least some kinds of exploration undergo nuanced changes across development [31, 32], and that the efficacy of interactive proprioceptive play may start improving as early as infancy–which may in turn be related to IQ scores later in development [33].

On the other hand, as they develop, children accumulate more experience with directed play. Older children's beliefs about the space of possible things to discover during directed play may thus be more constrained, leading to less variable exploration with age [34, 35]. Further, between four and six years of age, children in the US are typically exposed to increasingly formal learning environments (e.g., from daycares or preschools, to kindergartens or grade schools); therefore, the extent to which children's exploration is shaped by pedagogical contexts and demonstrations [2, 36] could shift during these years. Together, this past work suggests that patterns in exploratory play likely change with age, and that this may be related to later developmental outcomes. But *how* exactly directed play develops during the preschool years is not well understood.

## Current approach

Here we ask: How are patterns in children's exploratory play related to demographic factors and home environments in early childhood? Specifically, we investigate the extent to which age and SES account for variability in children's directed play. Because there is no way to run a true experiment when investigating effects of age and environmental contexts (which cannot be randomly assigned), here we compile data from several experiments already run in our lab and perform correlational analyses, in an attempt to preliminarily characterize these relationships. We designate this methodological approach a "mega-analysis": The increased sample size that results from compiling, normalizing, and analyzing across multiple studies may enable us to detect effects that would be difficult to discover in any of the individual studies' datasets. Indeed, the abundance of exploratory play data already collected within our lab, and the fact that our participants come from the highly socioeconomically diverse area around Newark NJ, conveniently situate us to use this approach.

## Method

Data for this analysis were compiled from four past experiments conducted within our lab between November 1 of 2014 through April 30 of 2019; these included a total of N = 278 children ($M_{(age)}$ = 56 months, range = 47–70 months, 150 girls), and represents 100% of the data collected in the lab that was suitable for the question of interest. All studies were approved by the Rutgers University–Newark Institutional Review Board (protocol numbers 14-144Mc, 16-

625Mc). Written consent was obtained from participants' parents or legal guardians for all experiments.

For all of the included experiments, children were recruited from and tested at various local preschools and daycare centers. The studies were run in relatively quiet locations within the testing sites (e.g., an empty teachers' lounge), to ensure that distractions from people or objects not involved in the study were kept to a minimum. Each experiment had three to four distinct experimental conditions built into their designs, investigated different research questions, and were run by different experimenters. What they had in common was their use of a novel toy exploration task in lightly scaffolded, "directed play" contexts, as all four experiments used some measures of play as their dependent variable of interest. In novel toy exploration tasks, children are presented with a unique toy that has several non-obvious causal affordances that could only be discovered through active engagement with the toy (e.g., a hidden button that makes the toy play music). In some cases, an experimenter calls the child's attention to a specific feature of the toy in the form of a pedagogical demonstration (e.g., "Look at this: You press this button to make my toy light up!") or pedagogical question (e.g., "I am asking you to think about: What does this button do?"). If a pedagogical cue is not used, the experimenter presents the toy to the child in a more neutral manner (e.g., "Look at this toy! You can play with it to figure out how it works!"). For the data included in this analysis, three of the experiments (N = 248) included a pedagogical demonstration, while one (N = 30) did not involve any kind of pedagogical cue.

Children are then allowed to play with and explore the toy until they tell the experimenter that they are done, or until an allotted amount of time has passed. These play sessions are video recorded, and trained researchers who are naïve to the hypotheses and conditions of the study code key aspects of children's play, including the amount of time children play with the toy, the number of functions they discover and how they focus on them, and the number of unique actions they perform on the toy.

## Playtime data

Because this analysis involved experiments that had already been fully collected, play data had been previously coded from videos. Previously coded play measures included:

- Total playtime: The total amount of time (in seconds) the child spent playing with the novel toy in some way. Available for 278/278 children.

- Key function playtime: The amount of time (in seconds) the child spent playing with the function that was pedagogically cued by the experimenter. Available for 248/278 cases (one experiment, N = 30, did not involve a pedagogical cue to a key function).

- Unique actions: A tally of the number of unique ways in which the child interacted with the novel toy throughout the course of their play. Available for 278/278 cases. Some experiments also separately coded the number of unique actions that occurred specifically within the first 60 seconds of playtime (available for 91/278 cases).

From these coded playtime measures we also computed two additional variables, typically investigated in exploratory play studies like these:

- Proportion of key function play: Key function playtime divided by total playtime.

- Variability rate: The total number of unique actions divided by the total playtime.

For each of these five variables, we first conducted outlier analyses, dropping data points that were more than 2 standard deviations away from the mean of each variable within each

experimental condition. (We ran all analyses reported below with outliers both included and excluded; there were no qualitative differences in the results.) This resulted in 20 data points being dropped–which, after accounting for the computed values that also had to be dropped as a result of these missing data, left us with: 266 cases for total playtime; 248 cases for key function playtime; 272 cases for total unique actions; 91 cases for first-minute unique actions; 236 cases for proportion of key function play; and 260 cases for variability rate.

**Marginalizing out role of condition.** One concern with the data is that different experimental manipulations would affect these different play measures (in fact, this was predicted by the original study designs). To control for this, after performing outlier analyses, we min-max normalized all values (i.e., re-scaled quantities to be on a scale from 0 to 1, where 0 represents the minimum, and 1 represents the maximum) for each variable, within each experimental condition. Marginalizing out all effects of condition and experiment prior to conducting any analyses ensured that any effects we saw in our data could not be attributed to differences in the studies themselves. All analyses reported here are conducted on the normalized data, as opposed to raw scores. See S1 Appendix for visualizations of each of these variables' normalized distributions, both overall and split by experiment.

### Demographic data

A key variable of interest was children's age. This variable was recorded for all participants in the study (278/278 children). In addition to children's age, we also had data for SES. For 199/278 children, we had access to families' home zip codes from consent forms. Using US census websites, we recorded the median income and the most common education level in the child's home zip code during the year in which the child was tested. Parents were provided the opportunity to report their individual income and education level when filling out consent forms, which would have been a more direct measure of the SES of individual children in our sample. However, very few parents opted to provide this information (N = 42). Encouragingly, median income by zip code correlated strongly with self-reported income within these 42 families ($r(40) = .615$, $p < .001$), suggesting that these group level measures are a reasonable estimate of individual differences in SES.

We also included analyses on children's gender. Some studies have suggested that preschool-aged boys may have better spatial reasoning abilities than girls [37, 38], which could lead to more efficiency when playing with a novel toy. On the other hand, other analyses have failed to find such gender differences until later in development [39] and our past work on play is consistent with this lack of difference. Nevertheless, given later achievement gaps between genders across various domains in classroom achievement and standardized tests [40, 41], we included gender in our analysis.

See S1 Appendix for visualizations of each of these variables' distributions, both overall and split by experiment. (N = 168 parents opted to provide information about their child's racial/ethnic background when filling out consent forms. For transparency, child race/ethnicity data is reported in S1 Appendix but is not included in any analyses).

De-identified data and analysis scripts are publicly available in our OSF repository for this project: https://osf.io/gdqnk/?view_only=95ad9948cdab43e8a44408e8b1be447d.

### Results

We examined the role of gender, age, and SES on the six (normalized) measures noted above (total playtime, key function playtime, unique actions–minute 1, unique actions–total, proportion of key function play, variability rate). We report only a subset of these analyses below, however all results are summarized in S2 Appendix. Consistent with prior work from our own

lab, no effect of gender was observed on any of the play measures and so this is not further reported or discussed as an additional factor.

### Child's age

The average age of children in our analysis was 56 months (4.67 years), ranging from 47–70 months (3.92–5.83 years), and with a standard deviation of 4.98 months.

We first ran correlations between age and all six play measures; to correct for familywise error, we set α to .008 for each of these analyses. We found that age was positively and significantly related to the total number of unique actions performed: $r(270) = .168$, $p = .005$ (see Fig 1). Thus, children's play tended to become more variable with age.

### Socioeconomic status

The average median income (per family) in our sample was $73,295.06, ranging from $25,192.00 to $198,625.00, and with a standard deviation of $39,161.26. Across zip codes, there were N = 125 cases for which "High School Graduate" was the most common education level,

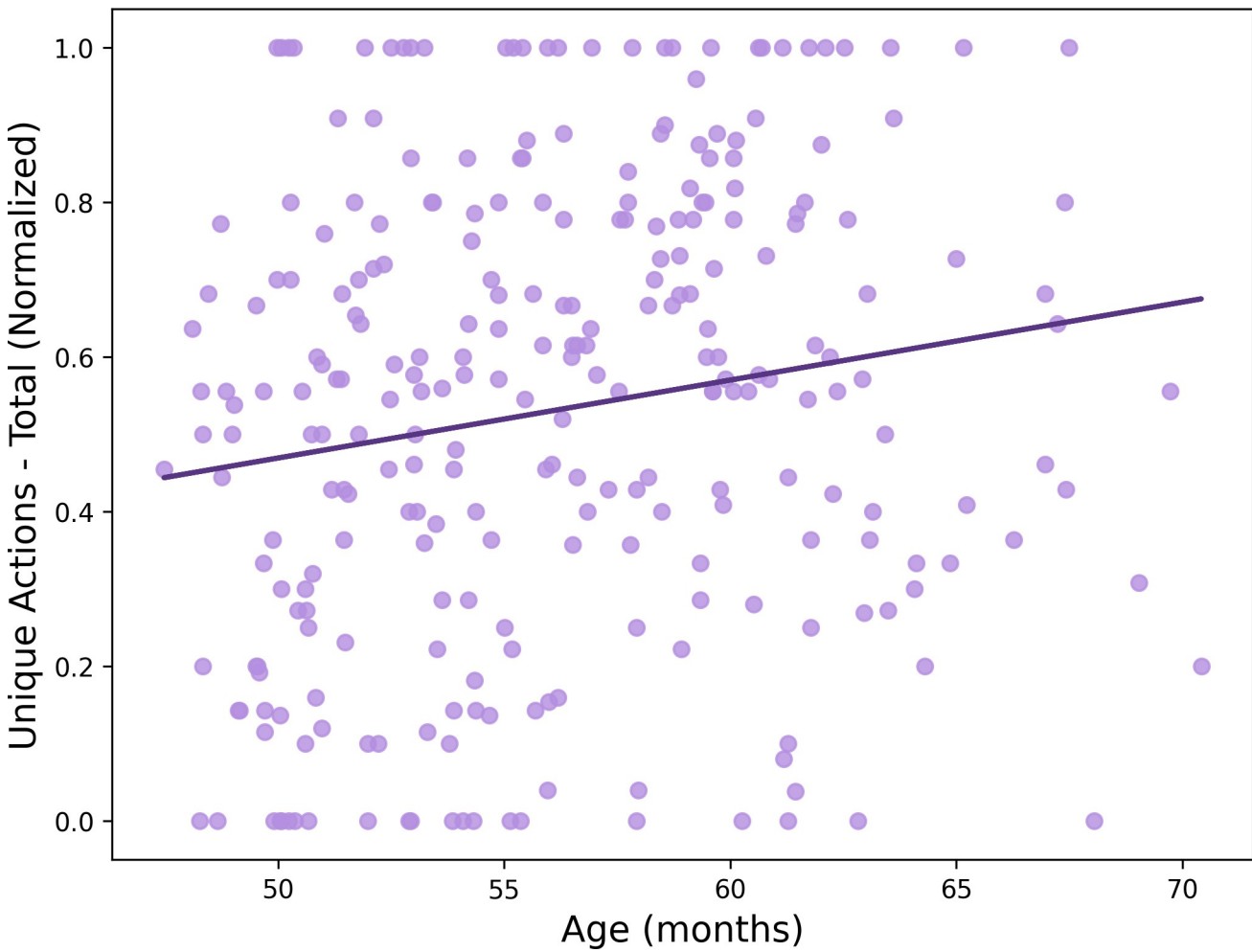

**Fig 1. Children perform more unique actions with age.** Children's age in months was positively correlated with the number of unique actions they performed on the novel toy (normalized), $p = .005$.

and N = 72 cases for which "Bachelor's Degree" was the most common education level. (There were also 2 cases for which "Graduate Degree" was the most common education level; for simplicity, we combined these with the "Bachelor's Degree" group for all subsequent analyses.) Income and education were tightly related across zip codes: $t(111.97) = 19.07$, $p < .001$ (unequal sample variance assumed). Children's age was not associated with either of these measures of SES ($ps \geq .148$). All children for whom we had SES data participated in experiments that involved some kind of pedagogical cue.

We correlated each of the six play measures with median income in the child's home zip code, and also ran t-tests using education level as the independent factor ($\alpha = .008$). Broadly, we found that higher socioeconomic status–particularly as captured by income–was associated with less variable play. Specifically, children from lower SES areas tended to perform more unique actions throughout the course of their play (income: $r(192) = -.193$, $p = .007$; education: $t(192) = 1.60$ $p = .112$; see Fig 2). Although children's age was not associated with our measures of SES, we also ran a partial correlation between income and unique actions while controlling for age, and found that the relationship remained significant ($r(192) = -.206$, $p = .004$).

How might children's tendency to focus on pedagogically demonstrated features potentially trade-off with the variability of their play, and how might this differ by SES? To address this question, we correlated the proportion of key function play with the total number of unique actions performed, split by education level ($\alpha = .025$). Within children from lower educated areas, the proportion of time spent playing with the key function traded off with overall variability–that is, the more of their time children spent focusing on what the experimenter demonstrated, the less variable their play tended to be ($r(119) = -.339$, $p < .001$). In contrast, we did not find analogous trade-offs in children from higher educated areas ($p = .58$; see Fig 3). A follow-up regression model predicting unique actions from education and proportion of key function play yielded a marginally significant interaction term ($p = .088$), so we remain cautious in our interpretation. Nevertheless, we tentatively report two take-aways: First, children from lower SES areas performed, on average, a more variable set of unique actions on the toy; and second, pedagogical focus could potentially trade off differently with variability of play depending on children's socioeconomic backgrounds.

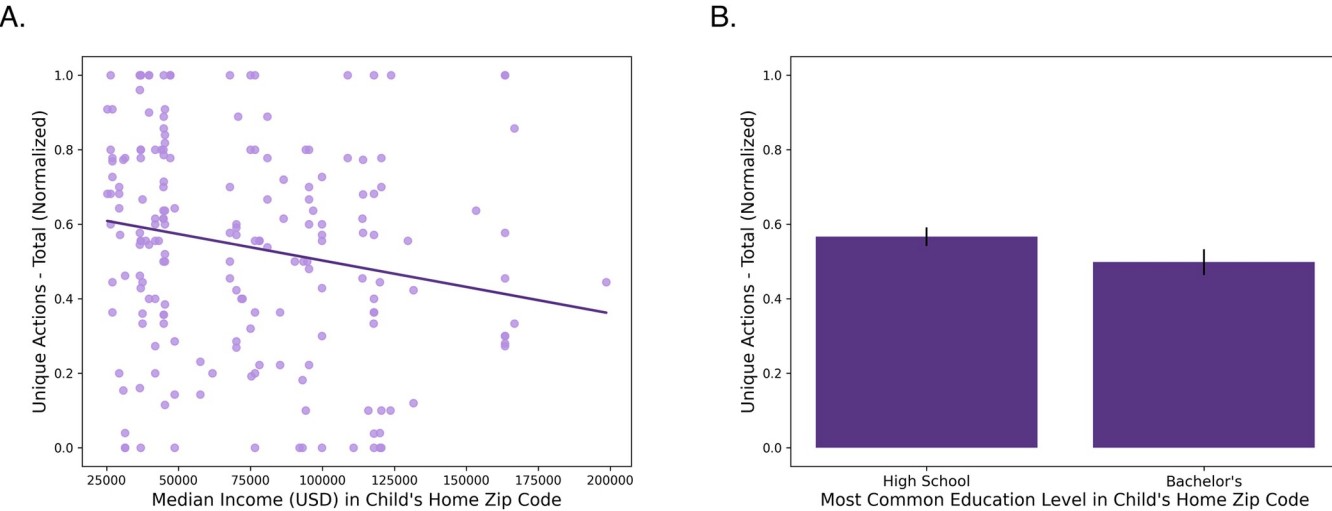

**Fig 2. Children from lower SES areas performed more unique actions.** Children who lived in lower SES areas performed more unique actions on the novel toy (normalized). SES was measured both by median income (A; $p = .007$) and modal education level (B; $p = .112$) in the child's home zip code.

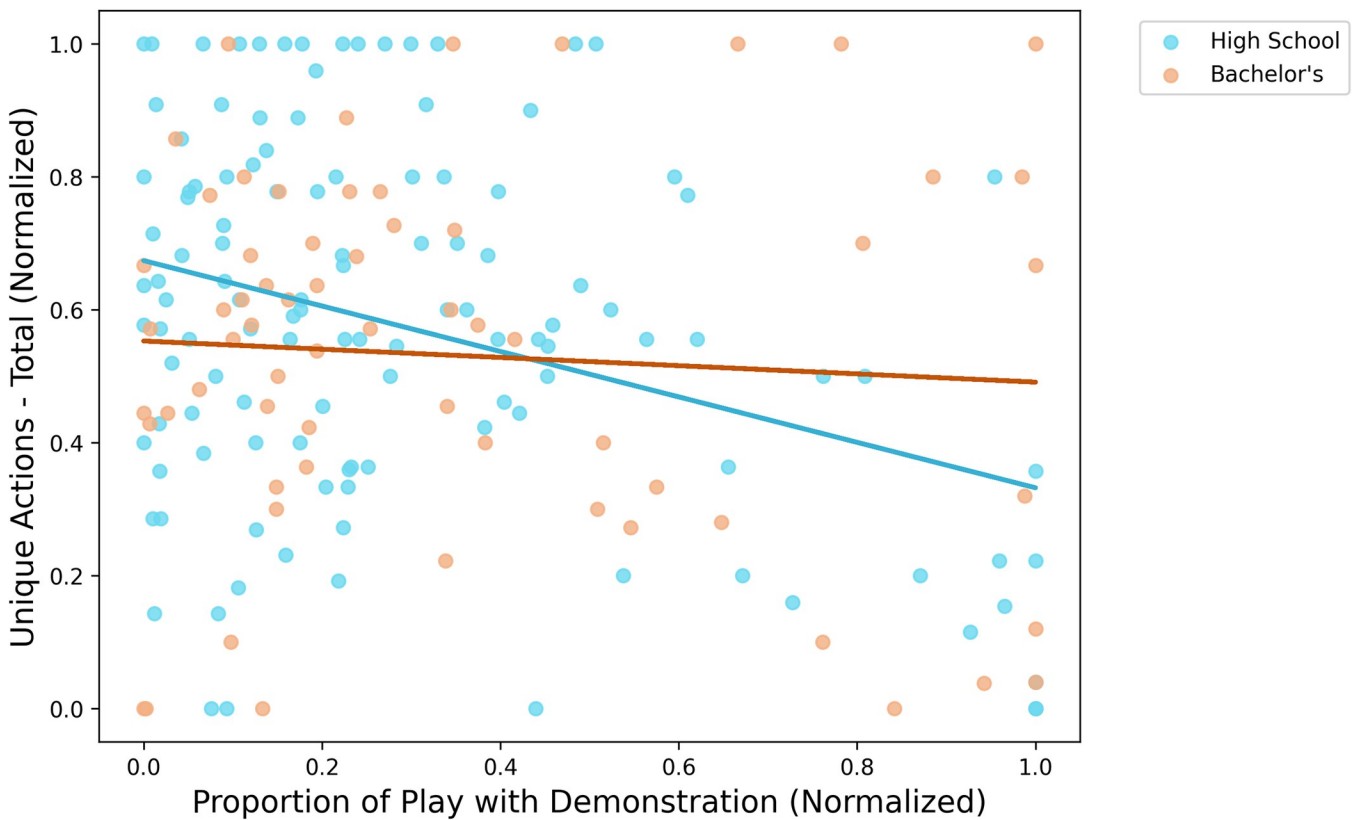

**Fig 3. Trade-offs between proportion of time spent playing with the demonstration and variability.** Within children from lower educated areas, the proportion of time spent playing with the demonstration traded off with overall variability of play ($p < .001$). We did not find such trade-offs in children from higher educated areas ($p = .58$).

## General discussion

The idea that children learn about the world through exploratory play is not a new one [15], but there is still much we do not know about exploration in early childhood. Recent work has demonstrated not only how children can leverage play to test hypotheses and resolve uncertainty [16, 19], but also some of the factors that may reasonably shift children's patterns of exploration at the group level [2, 3, 5, 22]. Less clear, however, is how individual differences are related to the ways in which children explore, particularly after observing pedagogical cues from adults. In this research, we compiled a large set of exploratory play data already collected and coded in our lab as a first step in characterizing the relationships between age, SES, and children's exploratory play. We report two key findings. First, as they get older, preschool-aged children tend to perform more unique actions on a novel toy. Second, children from lower SES areas tend to perform a more variable set of actions on the toy in general; additionally, the link between this play and a pedagogically-cued feature trades off differently than it does for higher SES children. Specifically, the tendency to focus on a pedagogically cued feature is associated with less variability of play for children from lower SES areas. In contrast, this trade-off does not exist for children from higher SES areas, as those children's tendency to focus on demonstrated features is not related to these other features of their play. This suggests that, for lower SES children, greater sensitivity to pedagogical cues also reflects a (reasonable) sensitivity to constrain exploration to only that demonstrated function, whereas for higher SES

children, initial interest in the pedagogical cue does not predict eventual exploration of other, non-cued features.

We found a developmental effect of exploration, such that preschoolers explored more variably with age. This is in line with past work suggesting that children's exploration becomes more directed and efficient as they develop [31], and suggests that something is indeed developing during these transformative years that broadly promotes exploration, which could in turn lend itself to broader discovery through play. On the other hand, it is also possible that age itself serves as a proxy for other potentially relevant constructs. For instance, children in these studies were all recruited from preschools and daycares, but they were as young as 3 years 11 months, and as old as 5 years 10 months. It could be the case that these older children had been enrolled in preschool for much longer than the younger children; thus, it would be difficult to disentangle the role of age from experience with these informal educational settings. Understanding the precise role of development in children's exploration is of course an open question for future work, but an exciting one nonetheless.

We also found that children from lower SES areas played more variably overall, and that this variability traded off with the degree to which low SES children focused on pedagogically demonstrated features. These observed differences may reflect different kinds of social expectations that children bring to bear on pedagogically cued events. Such expectations could be affected by myriad possible experiences that could differ as a function of environment. Given the rough proxy we used for SES in this analysis (and that SES itself served as a proxy for other environmental factors), there are a number of ways one could interpret these results. Perhaps children from lower SES areas are more motivated to explore, and potentially learn more from these tasks than children from higher SES environments. Maybe children from lower SES backgrounds have fewer or less variability of toys in their environments (either globally in their homes, or contextually in the daycares where they were tested), and so were more likely to capitalize on the opportunity to play with a novel toy. Further, it could be that (for children who were sensitive to the pedagogical prompt) lower SES children were more likely to interpret the pedagogical cues in our study as directives, while higher SES children saw them more as suggestions. This proposal is consistent with a host of research finding links between SES and parenting style, with more "permissive" styles of parenting increasing with SES [42, 43] (though see [44] for ways in which ethnicity may also play a role). Whatever the case may be, these SES-related results raise questions about how children may be differentially optimized for different environments [45]. However, what exactly those optimizations are, and how SES in particular relates to the experiences that serve as this causal mechanism, remains an open question. Together, we hope our results highlight how critical sample diversity is in understanding early learning in the developing mind.

We believe our "mega-analytic" approach of compiling and normalizing measures across multiple tasks can inform future work, as well. By capitalizing on data that had already been collected in the lab, we walk the line between empirical research in developmental psychology and meta-analysis. Our approach differs from a meta-analysis in that we were not limited to analyzing existing effect sizes–rather, we were actually able to conduct novel analyses on these data's dependent variables. At the same time, we were able to take a "big data" approach to deep questions about how children explore (or at least, bigger data than is typical for lab-based developmental research), providing us with more power and sample diversity than any of the individual studies would have had on their own. An additional timely advantage of our approach is that we did not have to collect any new data to conduct these new analyses–which, for interactive proprioceptive play-studies, has been more difficult during the COVID-19 pandemic. We hope others may build off our approach in future work.

There are a number of important limitations in our work, three of which we discuss here. First, as noted above, we used the median income in the child's home zip code as our primary measure of SES, which is a loose proxy for individual differences at the familial level. Moreover, we believe that SES is itself a proxy for meaningfully causal variables such as experience with different parenting styles [25], teaching styles [46], exposure to novel toys [34], resources in the daycare environment [11], and exposure to stress [13]. For example, we were not able to capture any qualities of the preschool or daycare environments that these children were attending when they participated in their respective studies. The degree to which such learning environments are more academically-focused versus play-based, for instance, could have influenced how children approached these novel toy tasks. Moreover, interactions with preschool teachers are some of the first formal pedagogical experiences children have. Qualities of these interactions could also reasonably affect how children interpret lightly scaffolded play moments from a new "teacher" (i.e., an experimenter), especially in the testing context of their preschool [46]. Of course, we used SES here out of convenience: It was the only previously collected measure that we had for most children across our studies that could in principle be tied to meaningful environmental variables. Understanding how exactly these early experiences coalesce to inform how children approach new opportunities for learning is a deep and fundamental question in child development, to be sure, and would make for exciting future work.

Second, we note the statistical limitations of this work. Because it is difficult to run true experiments when examining relationships between demographic factors, this work is correlational, and should be interpreted as such. We do not claim that SES or age *causes* the differences in variability we observed in children's play. Such studies would be challenging to explore experimentally, but are also simply unlikely under our theories of learning and play. Rather, we find it likely that the differential experiences children have through parenting and preschool exposure shape their expectations in our directed play tasks. Additionally, it is important to note that while we did correct for familywise error in our analyses, few of these analyses actually yielded significant results, calling into question the true sizes of these effects. Nevertheless, we are hopeful that the relationships we describe here are sufficiently meaningful to have raised new theoretical opportunities for investigation.

Finally, we acknowledge limitations in the generalizability of our results. Although our sample was relatively diverse in terms of SES and race/ethnicity, all of our participants were recruited in the United States; thus, we can likely only generalize these findings to WEIRD societies [47] at best. The development of children's exploratory play across different socioeconomic environments may well be couched in a larger cultural context that we were not able to examine here, though studies demonstrating pedagogical sensitivities on play have been extended to non-WEIRD populations, providing promise of potential generalizability [4]. Furthermore, we opted for high experimental control in these studies, which often trades off with the ecological validity of the results. To what extent does children's play behavior in these studies actually map onto playful learning and exploration in the messy environments of real life? Indeed, researchers have long struggled to even define "play" [48], which makes it a notoriously difficult behavior to study in development. Designing studies that opt for more ecological validity is one way in which future work could further connect the current findings to children's play in the real world.

In order to understand how children learn from others, and how they learn through play, we must understand how these factors interact. But what is more, we must also understand the variability that individual children bring to laboratory tasks as we build models of development. Home environments are likely tightly related to these sources of variability, but may be overlooked in traditional lab-based developmental science, as children's data are often analyzed in the aggregate. Our results suggest important points of individual difference borne

from different home and early school experience that could be meaningfully related to the development of children's playful exploration and pedagogical sensitivity during the preschool years. In future work, we hope researchers will continue to embrace the variability in their samples as a possible feature to explore, ultimately leading to a more complete picture of child development.

## Supporting information

**S1 Appendix. Full frequency distributions for each of the variables in our dataset, split by experiment.**
(PDF)

**S2 Appendix. A summary of all of the analyses performed on our compiled dataset.**
(PDF)

## Acknowledgments

Thank you to Amanda Castro, Joseph Colantonio, Jack Fredricks, Zachary Walden, and Yue Yu, who led the original studies that were included in this paper. De-identified data and analysis scripts are publicly available in our OSF repository for this project: https://osf.io/gdqnk/?view_only=95ad9948cdab43e8a44408e8b1be447d.

## Author Contributions

**Conceptualization:** Ilona Bass, Elizabeth Bonawitz.

**Data curation:** Ilona Bass.

**Formal analysis:** Ilona Bass.

**Funding acquisition:** Ilona Bass, Elizabeth Bonawitz.

**Investigation:** Ilona Bass.

**Methodology:** Ilona Bass, Elizabeth Bonawitz.

**Project administration:** Ilona Bass, Elizabeth Bonawitz.

**Resources:** Ilona Bass, Elizabeth Bonawitz.

**Supervision:** Elizabeth Bonawitz.

**Visualization:** Ilona Bass.

**Writing – original draft:** Ilona Bass, Elizabeth Bonawitz.

**Writing – review & editing:** Ilona Bass, Elizabeth Bonawitz.

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
