## [Decision Letter · Decision Letter 0]

6 Mar 2024

PONE-D-23-26158Early Environments and Exploration in the Preschool YearsPLOS ONE

Dear Dr. Bass,

Thank you for submitting your manuscript to PLOS ONE. After careful consideration, we feel that it has merit but does not fully meet PLOS ONE’s publication criteria as it currently stands. Therefore, we invite you to submit a revised version of the manuscript that addresses the points raised during the review process. Please submit your revised manuscript by Apr 20 2024 11:59PM. If you will need more time than this to complete your revisions, please reply to this message or contact the journal office at plosone@plos.org. Sorry to have taken so long to return this piece, I struggled to find a second reviewer, so I served in that capacity. The article is easy to read and clearly explains how you used extant data to answer a new question. The reviewer included has some comments and issues for you to address.  

Please include the following items when submitting your revised manuscript:A rebuttal letter that responds to each point raised by the academic editor and reviewer(s). You should upload this letter as a separate file labeled 'Response to Reviewers'.A marked-up copy of your manuscript that highlights changes made to the original version. You should upload this as a separate file labeled 'Revised Manuscript with Track Changes'.An unmarked version of your revised paper without tracked changes. You should upload this as a separate file labeled 'Manuscript'.If applicable, we recommend that you deposit your laboratory protocols in protocols.io to enhance the reproducibility of your results. Protocols.io assigns your protocol its own identifier (DOI) so that it can be cited independently in the future. For instructions see: https://journals.plos.org/plosone/s/submission-guidelines#loc-laboratory-protocols. Additionally, PLOS ONE offers an option for publishing peer-reviewed Lab Protocol articles, which describe protocols hosted on protocols.io. Read more information on sharing protocols at https://plos.org/protocols?utm_medium=editorial-email&utm_source=authorletters&utm_campaign=protocols.

We look forward to receiving your revised manuscript.

Kind regards,

Mary Diane Clark, PhD

Academic Editor

PLOS ONE

Journal Requirements:

“This work was funded in part by an Early Career Research Fellowship from the Jacobs Foundation to EB (https://jacobsfoundation.org/), and a Scholar Award in Understanding Human Cognition from the James S. McDonnell Foundation to EB (https://www.jsmf.org/). IB is supported by a Mind Brain Behavior Postdoctoral Fellowship from Harvard University (https://mbb.harvard.edu/). The funders had no role in study design, data collection and analysis, decision to publish, or preparation of the manuscript.”

4. Please update your submission to use the PLOS LaTeX template. The template and more information on our requirements for LaTeX submissions can be found at http://journals.plos.org/plosone/s/latex.

Additional Editor Comments (if provided):

Nicely done article, thank you for allow us to review it. I have only one review for you as I was unable to obtain a second review. This new analysis of your past data collection shows some potentially important information that can be. used in many areas, including. Education, Recreation, and Parental Interventions.

I note one thing that I will ask you to change. You have a don't at the top of your Discussion section.

Then please respond to the comments from Reviewer !.

Reviewers' comments:

Reviewer's Responses to Questions

**Comments to the Author**

1. Is the manuscript technically sound, and do the data support the conclusions?

Reviewer #1: Partly

2. Has the statistical analysis been performed appropriately and rigorously? 

Reviewer #1: Yes

3. Have the authors made all data underlying the findings in their manuscript fully available?

Reviewer #1: Yes

4. Is the manuscript presented in an intelligible fashion and written in standard English?

Reviewer #1: Yes

5. Review Comments to the Author

Reviewer #1: This manuscripts presents a relatively novel analysis that helps researchers address differences in methodologies and still explore underlying patterns in the larger data sets.

The focus of the study, development of exploratory play behavior is valuable both for understanding cognitive development and for educational and curricular applications.

There are, however, some issues to address in the manuscript. Addressing these issues would help to contextualize the results and elucidate more clearly the next steps that need to be explored.

1. The methods section notes that in some experiments children received pedagogical demonstration and in others a pedagogical direction - did this occur in all 4 experiments, and if not how was the toy introduced?

2. More information on the context of the experiments would be useful - how many people in the room, were there other toys or objects in the room, etc.

3. The age range was just shy of age 4 to just shy of age 6, were some of the 5-year-olds in Kindergarten, Transitional Kindergarten, or PreK? While you note that age differences could also reflect exposure to formal school, this point is under-emphasized. If not, a central question is whether more formal school experience informs exploratory play behavior that differ from more informal daycare or preschool settings. (what were the preschool/day care settings like - were they play based, academic, etc.)

4. I recognize that you wanted to control for variations in methodology across the different experiments, but did children's exploratory play behavior differ by prompt, controlling for age and income?

5. What are the correlations among the variations of interest?

6. The discussion section needs to acknowledge that you only had 2 of 24 correlations be statistically significant;

7. The discussion section needs to more strongly emphasize that results in lab-like settings do not necessarily reflect messy everyday life, where children likely are not often handed a single toy to explore in a relatively (I assume) empty and quiet room.

8. Some discussion of how and when exploratory play

6. PLOS authors have the option to publish the peer review history of their article (what does this mean?). If published, this will include your full peer review and any attached files.

Reviewer #1: No

---

## [Author Response · Author response to Decision Letter 0]

17 Apr 2024

We are very grateful for the opportunity to revise our manuscript. We have read carefully through the comments and have made changes throughout the manuscript to address the feedback. We believe these changes have significantly improved the paper. Please see our full Response to Reviewers for a point-by-point response. Thank you again for the opportunity.

---

## [Editor Report · Decision Letter 1]

29 May 2024

Early Environments and Exploration in the Preschool Years

PONE-D-23-26158R1

Dear Dr. Bass,

We’re pleased to inform you that your manuscript has been judged scientifically suitable for publication and will be formally accepted for publication once it meets all outstanding technical requirements.

Kind regards,

Mary Diane Clark, PhD

Academic Editor

PLOS ONE

Additional Editor Comments (optional):

I jumped in to do a review as I could only get one. The other reviewer and I both agree you only need a minor reviision. I look forward to your revision.
---

## [Editor Report · Acceptance letter]

30 May 2024

PONE-D-23-26158R1 

PLOS ONE

Dear Dr. Bass, 

I'm pleased to inform you that your manuscript has been deemed suitable for publication in PLOS ONE. Congratulations! Your manuscript is now being handed over to our production team.

Kind regards, 

on behalf of

Dr. Mary Diane Clark 

Academic Editor

PLOS ONE